# Observation of Boyer-Wolf Gaussian modes

Konrad Tschernig[1,2], David Guacaneme [1,2], Oussama Mhibik[1], Ivan Divliansky[1] & Miguel A. Bandres [1] ✉

Stable laser resonators support three fundamental families of transverse modes: the Hermite, Laguerre, and Ince Gaussian modes. These modes are crucial for understanding complex resonators, beam propagation, and structured light. We experimentally observe a new family of fundamental laser modes in stable resonators: Boyer-Wolf Gaussian modes. By studying the isomorphism between laser cavities and quadratic Hamiltonians, we design a laser resonator equivalent to a quantum two-dimensional anisotropic harmonic oscillator with a 2:1 frequency ratio. The generated Boyer-Wolf Gaussian modes exhibit a parabolic structure and show remarkable agreement with our theoretical predictions. These modes are also eigenmodes of a 2:1 anisotropic gradient refractive index medium, suggesting their presence in any physical system with a 2:1 anisotropic quadratic potential. We identify a transition connecting Boyer-Wolf Gaussian modes to Weber nondiffractive parabolic beams. These new modes are foundational for structured light, and open exciting possibilities for applications in laser micromachining, particle micromanipulation, and optical communications.

The laser is one of the greatest inventions in optics and photonics. Lasers drive modern research in physics, chemistry, and biology; they are a vital tool in manufacturing and medicine, and the heart of Internet communications. Everything started with the first optical laser resonator by Maiman[1], where a ruby crystal excited by a flash lamp pump generates a resonant spatial mode within a planar-planar cavity. From there, the laser has exponentially evolved from macroscopic spherical mirror resonator cavities to microcavities, fiber, and semiconductor lasers that are present in almost any modern electronic device. Although the miniaturization of lasers is vital for technology, stable macroscopic resonators, where light is confined by a quadratic phase, remain highly relevant in terms of high power, stability, beam quality, linewidth, and spectral control.

Stable resonator cavities can support, apart from the fundamental Gaussian beam, three different fundamental families of transverse modes: the Hermite-Gaussian modes (HGM) in Cartesian coordinates, Laguerre Gaussian modes (LGM)[2] in cylindrical coordinates, and Ince Gaussian modes (IGM)[3,4] in elliptical coordinates. These families of modes are important not only in optics and photonics but also in physics and mathematical physics. From the physics point of view, the modes of stable laser resonators are equivalent to the eigenstates of a

2D quantum harmonic oscillator, i.e., the round trip in the optical cavity has the same Hamiltonian, with the parabolic cavity mirrors acting like a quadratic potential. In this regard, the modes of the stable laser resonator resemble the probability distribution of quantum particles in a harmonic potential, which is an essential cornerstone in the study of quantum mechanics as it allows us to investigate more complicated potentials[5]. For example, by using a nonplanar twisted cavity, which induces a "magnetic" coupling between the modes of a fundamental laser cavity, it has been possible to observe synthetic Landau levels for photons[6] and Laughlin states made of light[7].

From the mathematical physics side, the study of integrable and superintegrable Hamiltonians, like the cavity Hamiltonian, is the key to understanding symmetries and solutions of differential equations. Indeed, the reason why a stable resonator cavity with spherical mirrors can only support three separable families of modes is tied to the mathematical fact that the cavity Hamiltonian only commutes with three different second-order symmetry operators[8–10], each one associated with a different coordinate system and a different physical conserved quantity, i.e., the LGM in cylindrical coordinates with conserved angular momentum. Although groups of modes of the stable laser cavity are degenerate in terms of the resonant frequency, they are

[1]CREOL, The College of Optics and Photonics, The University of Central Florida, Orlando, FL, USA. [2]These authors contributed equally: Konrad Tschernig, David Guacaneme. ✉e-mail: bandres@creol.ucf.edu

not degenerate with respect to the second-order eigenoperator that commutes with the cavity Hamiltonian. This explains why the cavity generally lases in a single mode of a given family, LGM, HGM, or IGM, and not in a superposition of a degenerate frequency group within each family[11–13],–where any small misalignment of the cavity dictates which family is more favorable to lase.

From the perspective of optics and photonics, the three fundamental families of modes of spherical stable resonators, the HGM, LGM, and IGM, are of paramount significance. Besides being the modes of the most elemental laser cavity, they are key in our understanding of more complex cavities[14–16], beam propagation, and most importantly, they serve as a cornerstone of structured light. Over the last decade, our ability to tailor optical fields, both spatially and temporally, has significantly advanced. These advances have been driven by both fundamental science as well as engineering applications. Beams that carry orbital angular momentum[17], as the Laguerre Gauss modes are perhaps the most well-known example of structured light[18,19]. Two main families of structured light, the nondiffractive beams[20–22], and the accelerating beams[23–25], have flourished in applications during the last decade, from biomedical imaging[26] and free space communications[27–30], to micromanipulation[31–34] and micromachining[35–38]. To understand how the fundamental families of Gaussian beams are a cornerstone of structured light, we should examine closely the most recent addition to the known families of fundamental modes of stable laser resonators, the Ince Gaussian beams. The IGMs constitute the third complete family of transverse modes of stable resonators, they were theoretically introduced[3] and experimentally observed[4] in 2004. Due to their inherent elliptical symmetry, they form a complete family of resonator modes for each value of the focal distance of the elliptical coordinate system. For this reason, although at the beginning they were thought of as an elusive solution, it was quickly realized that they are more common in any resonator than the HGM and LGM[4]. Indeed, the LGMs and HGMs

correspond to limiting cases of the IGMs when the focal distances tend to zero or to infinity, respectively, as depicted in Fig. 1. IGMs have been generated in many different laser architectures and have quickly found a plethora of applications such as manipulation of microparticles[39], nonlinear two-wave mixing[40], microlaser resonators[41], quantum angular momentum[42], quantum entanglement[43], and single-cell biological lasers[44], among many others.

Notably, in a similar way that an IGM can transition to an HGM or LGM mode by changing the focal distance, these three families of fundamental cavity modes can transform into nondiffractive beams or accelerating beams by altering their mode number and waist in specific ways[45–47]. These transitions are clearly depicted in Fig. 1. In this way, one can generate modes that closely resemble nondiffractive or accelerating beams directly from a spherical stable resonator under the right conditions. Considering the significance of the fundamental families of modes in spherical laser resonators, it is intriguing to consider the possibility that a new family of lasing modes exists.

## Results

Here, we present the experimental observation of a new family of fundamental laser modes of stable resonators: the Boyer-Wolf Gaussian modes. By studying the isomorphism between stable laser resonator cavities and quadratic Hamiltonians, we designed a laser resonator equivalent to a quantum two-dimensional anisotropic harmonic oscillator with a 2:1 frequency ratio[48]. The Boyer-Wolf Gaussian modes emerge in a parabolic coordinate system and are constructed by a product of solutions to the sextic anharmonic oscillator. Due to their inherent parabolic symmetry the Boyer-Wolf Gaussian modes break the symmetry around the $y$-axis that is present in the HGM, LGM, and IGM, and create a dark parabolic region around the $x$-axis. We observed the first 18 Boyer-Wolf Gaussian modes lasing from our cylindrical lens resonator; the experimental results exhibit an excellent agreement with the theoretical predictions. Furthermore, we show

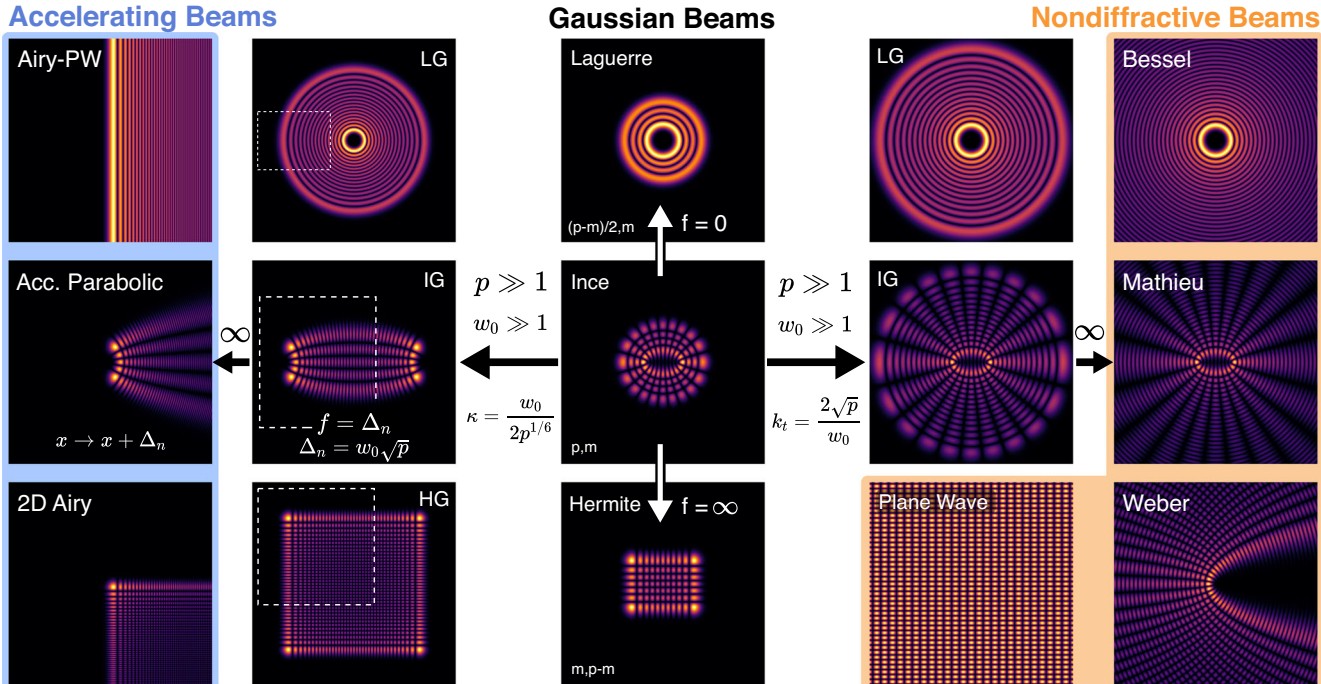

**Fig. 1 | Fundamental modes of stable laser resonators.** *Middle panel*, the three families of modes of spherical stable laser resonators: Laguerre-Gaussian, Ince-Gaussian, and Hermite-Gaussian beams. The Ince-Gaussian modes converge to the Laguerre-Gaussian and Hermite-Gaussian when the focal distance tends to zero and infinity respectively. *Left panel*, the transition of the Gaussian beams to accelerating beams. The transformation occurs as the order, $p$, and the waist size, $w_0$, go to

infinity while keeping the ratio $\kappa = w_0/p^{1/6}$, which becomes the transverse scale of the accelerating beam, constant, and by shifting the coordinates to the left peak of the beam. *Right panel*, transmutation of the Gaussian beams to nondiffracting beams. The metamorphosis takes place when the order, $p$, and the waist size, $w_0$, tend towards infinity, while maintaining the ratio $k_t = 2\sqrt{p}/w_0$ constant, which becomes the transverse wavevector of the nondiffractive beam.

that the Boyer-Wolf Gaussian modes are also eigenmodes of an anisotropic gradient refractive index medium with a 2:1 ratio between the parabolic index of refraction profile at each axis. This implies that the Boyer-Wolf modes must also be present in any physical system—such as ultracold atoms, mechanical and acoustic systems—with a 2:1 anisotropic quadratic potential. Finally, we find a transition that connects the Boyer-Wolf Gaussian modes to the Weber nondiffractive parabolic beams[21], showing that Boyer-Wolf modes are a foundational element of structured light. The new Boyer-Wolf Gaussian modes demonstrated here open the door to new applications in laser micromachining, particle micromanipulation, and optical communications.

## Theory

We begin by analyzing the duality between rays and wave optics in optical resonators. As is well known, a paraxial cavity can be analyzed based on its total ABCD matrix, which describes the complete round trip evolution of all light rays in the resonator. This "geometrical" ray analysis unfolds many important properties of the cavity, such as its stability structure[2]. To find the transverse modes, one needs to find the beams that self-reproduce after propagating a round trip in the laser resonator. This propagation is described by the Collins diffraction integral given by the ABCD matrix of the resonator, and the eigenmodes of such integral operator are the modes of the cavity. Finding the eigenmodes of such linear integral operators is an analytically challenging problem; therefore, this approach is usually limited to numerical simulations[49].

Recently, a new formalism has shown that by considering the periodicity of a round trip in the resonator, it is possible to find a Floquet Hamiltonian that describes the transverse cavity modes[50]. More importantly, the parameters of this transverse quadratic Hamiltonian only depend on the total ABCD matrix of the resonator, and solutions for such Hamiltonians have been studied extensively in mathematics[9,10,51]. This outcome establishes a significant link between the ray optics and wave optics within an optical laser resonator. Following this formalism, one can show that the transverse modes of stable spherical resonators are eigenmodes of the quantum two-dimensional isotropic harmonic oscillator, $V(x,y) = \kappa^2(x^2 + y^2)$, where $\kappa$ is given in terms of the parameters of the resonator, see Supplementary Note 1. From the theory of separation of variables, it is known that such Hamiltonians are only separable in cartesian, circular and elliptical coordinates[10]. For this reason, spherical optical cavities only support three separable families of fundamental modes, the Hermite, Laguerre and Ince-Gaussian modes[3,4].

We emphasize that the primary limitation of this formalism[50] is that the theoretical description of ABCD systems only considers "quadratic optics" and does not include higher-order aberrations, such as mirror imperfections and non-paraxial corrections. Here, and generally, the paraxial approximation is entirely adequate because deviations from this approximation are usually small when the resonator mode waist $w_0$ is significantly larger than the wavelength $\lambda$. Therefore, for our paraxial resonator, these corrections only become important for misalignments outside the paraxial regime, for very higher-order modes, and for resonator configurations with very small waist. A complete theory of resonator aberrations for nonparaxial beam propagation and optical elements beyond-quadratic mirrors and lenses can be found in ref. 52.

To find new modes of a laser resonator, we begin by identifying quantum Hamiltonians with quadratic potentials that are integrable, meaning they have analytic solutions in a given coordinate system. Such Hamiltonians have been extensively studied and classified in mathematics in the area of integrability and superintegrability[8–10,51]. The simplest option to introduce a new potential is to promote the quantum harmonic oscillator Hamiltonian, which describes spherical resonators, to a quantum anisotropic harmonic one, where the $x$ and $y$ axis parabolic potential have different strengths,

$V(x,y) = \kappa_x^2 x^2 + \kappa_y^2 y^2, \kappa_x \neq \kappa_y$. However, the solutions of such a potential are a product of two orthogonal Hermite-Gaussian beams with different waist sizes, bringing us back to the known families of modes. Here it is important to point out that, although any quadratic Hamiltonian can be mapped to an ABCD matrix, not every ABCD matrix can represent the round-trip propagation of an optical cavity with a simple configuration and conventional optical elements. For instance, if we implement the anisotropic oscillator using an astigmatic spherical mirror, the resonator will only exhibit the desired ratio $\kappa_x : \kappa_y$, for one specific cavity size (see Supplementary Note 2).

Interestingly, the quantum two-dimensional anisotropic harmonic oscillator with a 2:1 frequency ratio is separable in parabolic coordinates[9]. The corresponding time independent Schrödinger equation is given by

$$\hat{H}\psi = -\left(\partial_x^2 + \partial_y^2\right)\psi + \kappa^2\left(4x^2 + y^2\right)\psi = E\psi, \tag{1}$$

where $\kappa$ is the "strength" of the potential and $E$ is the "eigenenergy" of the mode. From the optics point of view, as we will explain below, $\kappa$ is given in terms of the parameters of the resonator and the lasing wavelength and it characterizes the waist of the lasing modes; while $E$, is proportional to the resonant frequency which is related to the Gouy-shift phase—the total phase accumulation during a round-trip in the resonator. Boyer and Wolf studied the separation of variables, solutions, and symmetries of Eq. (1) extensively in[48]. The solution to Eq. (1) in parabolic coordinates $(u,v)$, $x = (u^2 - v^2)/2$, $y = uv$, $u \in \mathbb{R}$, $v \in \mathbb{R}^+$ is given by, $BWG_{nl}(u,v) = \phi_{nl}(u)\phi_{nl}(iv)$,

$$\mathrm{BWG}_{nl}(u,v) = c_{nl}\left(\frac{\sqrt{2}uv}{w_0}\right)^n \exp\left(-\frac{u^4 + v^4}{2w_0^2}\right) P_{nl}\left(\left[4\frac{u^2}{w_0}\right]^{-1}\right) P_{nl}\left(-\left[4\frac{v^2}{w_0}\right]^{-1}\right), \tag{2}$$

where $w_0^2 = 2/\kappa$ is the beam waist, and $c_{nl}$ is a normalization constant, $\phi_{nl}(\bullet)$ are solutions to the sextic anharmonic oscillator, and $P_{nl}(\bullet)$ is a "parabolic" polynomial of degree $\lfloor n/2 \rfloor$, see Supplementary Notes 2, 3, and 5 for more details. The modes are characterized by the mode numbers $n = 0,1,2,\ldots$, and $|l| \leq 1/2\lfloor n/2 \rfloor$. We will call these solutions the Boyer-Wolf Gaussian modes (BWG). In Fig. 2 we show the theoretical intensity pattern of the Boyer-Wolf Gaussian modes, where their parabolic nature is clearly depicted by their parabolic nodal lines. The modes with the same order $n = 0,1,2,\ldots$, are $\lfloor n/2 \rfloor + 1$ degenerate, and for this reason they will have the same resonant frequency and Gouy phase shift. In contrast to the fundamental families, HGM, LGM, IGM, the BWG modes break the symmetry around the $y$-axis and create a dark parabolic region around the $x$-axis.

## Experiments

The next important task is to find an optical resonator whose round trip ABCD matrix could be mapped to the quantum two-dimensional anisotropic harmonic oscillator with a 2:1 frequency ratio, as given in Eq. (1). Consequently, the resonator will generate the BWG modes inside the cavity. To achieve this, we work backwards, given Eq. (1), we find its corresponding total ABCD matrix, and afterwards a cavity that is described by such a ray transfer matrix. One could find few options of such systems; however, they usually have major drawbacks. Firstly, they require unconventional optical elements such astigmatic mirrors, which are quite challenging to manufacture. Secondly, the resonators usually only exhibit the desired 2:1 ratio for one specific distance of the mirrors, which further complicates the assembly of such a resonator. We found an elegant solution that overcome these constraints by exploiting the symmetries of the equation as described in the Supplementary Note 2. Our BW resonator, depicted in Fig. 3a, uses a cylindrical lens with focal distances $f_x$ in the $x$-axis, and a cylindrical mirror with focal distance $f_y = 2f_x$, in the $y$-axis. The $y$-axis cylindrical

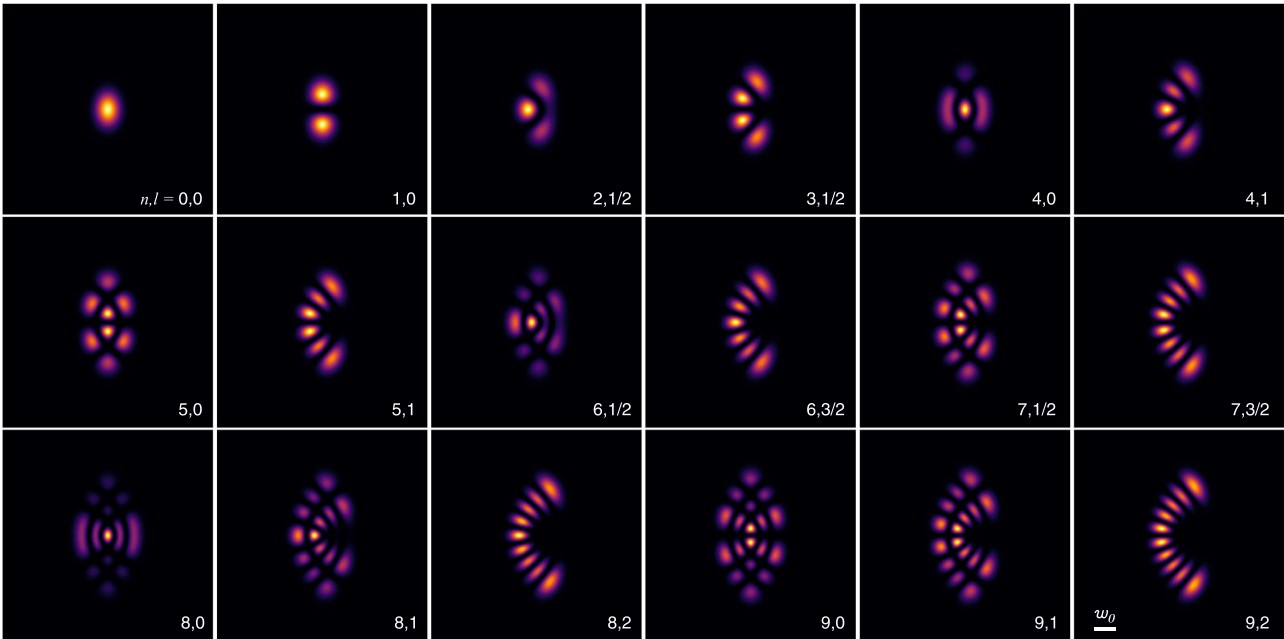

**Fig. 2 | Theoretical Boyer-Wolf Gaussian modes.** Beam intensity pattern of the first eighteen Boyer-Wolf Gaussian modes of the 2:1 anisotropic laser cavity. The mode order and degree ($n,l$) are displayed in the bottom right corner of each panel. Modes with the same order $n = 0,1,2,\ldots$, have the same resonating frequency and Gouy-shift phase. Due to the parabolic nature of the modes, the nodal lines form parabolas. The parity of $n$ indicates the symmetry of the mode with respect to the $y$-axis. Modes with negative $l$ are a reflection around the $y$-axis ($x \rightarrow -x$) of the positive $l$-modes depicted here.

mirror, $f_y = 2f_x$, together with the output coupler flat mirror, creates a 1D cavity of length $L$ in the $y$-axis—this leads to a harmonic oscillator potential in the $y$-direction with $\kappa_y = \kappa = k[L(2f_y - L)]^{-1/2}$, where $k = 2\pi/\lambda$, and $\lambda$ is the wavelength. Now, the crucial step is to place an $x$-axis cylindrical lens, $f_x$, in the middle of the cavity at a distance, $L_x = L/2$, as shown in Fig. 3a. Considering the equivalent unfolded resonator, note that one round trip of the whole resonator, corresponds to one Fabry-Pérot resonator of length $L$ in the $y$-axis, but two resonators of length $L/2$ in the $x$-axis. As a result, this leads to a harmonic oscillator potential in the $x$-direction with $\kappa_x = k[L/2(2f_x - L/2)]^{-1/2} = 2\kappa_y$ – achieving the desired ratio $\kappa_x : \kappa_y = 2:1$ of the two-dimensional anisotropic harmonic oscillator. The advantage of this cavity design is that for any pair of cylindrical lenses with $f_y = 2f_x$, the resonator is stable for any $L \leq 4f_x$. The waist of the modes at the flat mirror output coupler, where the phase front is plane, is $w_0^2 = 2k^{-1}\sqrt{L(2f_y - L)}$. The resonant condition depends on the "energy" of the modes, that is given by the mode number $n$. The resonance occurs when the phase shift for a round trip in the cavity is a multiple of $\pi$. As a result, the transverse mode frequency spacing between consecutive Boyer-Wolf modes is given by

$$\Delta v = \left(\frac{c}{2L}\right)\frac{1}{2\pi}\left(n + 3/2\right)\arccos\left(1 - L/R_x\right), \qquad (3)$$

where $c$ is the velocity of the light. We highlight further properties of the BWG modes and the 2:1 cavity in the Supplementary Notes 4 and 6.

To generate the Boyer-Wolf Gaussian modes we build the solid-state laser cavity depicted in Fig. 3a. Our laser resonator consists of an Nd:YAG crystal that acts as a gain medium and output coupler, an $x$-axis cylindrical lens, $f_x = 100mm$, at the middle of the cavity and a $y$-axis cylindrical mirror, $f_y = 200mm$, at the end of the cavity. We used different resonator lengths from $L = 12cm$ to $30cm$. The laser is optically pumped by a high-power laser diode (808 nm) that is focused on the Nd:YAG crystal which has a high transmission at the diode

wavelength (808 nm) and high reflectance at the lasing wavelength (1064 nm). All details about the laser resonator are given in the Methods section. The perfectly aligned cavity quickly lases in the fundamental Boyer-Wolf Gaussian mode—an elliptical Gaussian mode with waist ratio $\sqrt{2} : 1$. To generate higher-order Boyer-Wolf Gaussian modes, we slightly misaligned the cavity in different ways. This was achieved by adjusting the cavity length and shifting the position of the optical axis relative to the pump beam, through slight tilting and displacement of the cylindrical mirror and lens, see Supplementary Notes 9, 10. We observed the first eighteen Boyer-Wolf Gaussian modes lasing in the cavity—our experimental results are shown in Fig. 3b. The observed modes clearly show their characteristic parabolic nature, with mode profiles that are markedly distinct from those of the fundamental HGM, LGM, and IGM modes. All the modes have a waist size that aligns with the parameters of the resonator, and the higher order $n$ modes have larger extents than those with lower $n$ indices, as expected. We emphasize the excellent agreement between the measured and theoretical patterns shown in Fig. 3b and Fig. 2.

The full round-trip propagation of the Boyer-Wolf Gaussian modes inside the "unfolded" laser resonator is simulated and depicted in Fig. 4a. From the $x$ and $y$ cross-sections of the propagation, it is clear that the cavity behaves like a single resonator of length $L$ along the $y$-axis, while along the $x$-axis, it behaves like two resonators, each of length $L/2$. Note that the Bower-Wolf Gaussian modes are not scale-invariant under propagation due to their 2:1 anisotropic nature. We study their free-space propagation as well as their Fourier transform, i.e., far field, in detail in the Supplementary Note 4.

## The 2:1 GRIN medium

Interestingly, there is another optical system whose paraxial wave equation is also described by Eq. (1); this is a 2:1 anisotropic gradient index medium (GRIN). This situation parallels the cases of the fundamental HGM, LGM, and IGM, which are not only modes of a stable spherical resonator but also eigenmodes of an isotropic GRIN medium[53]. The index of refraction of a 2:1 anisotropic GRIN medium is given by $n^2 = n_0^2\left[1 - a^2\left((2x)^2 + y^2\right)\right]$, where $n_0$ is the reference

**a** Anisotropic 2:1 Laser Cavity

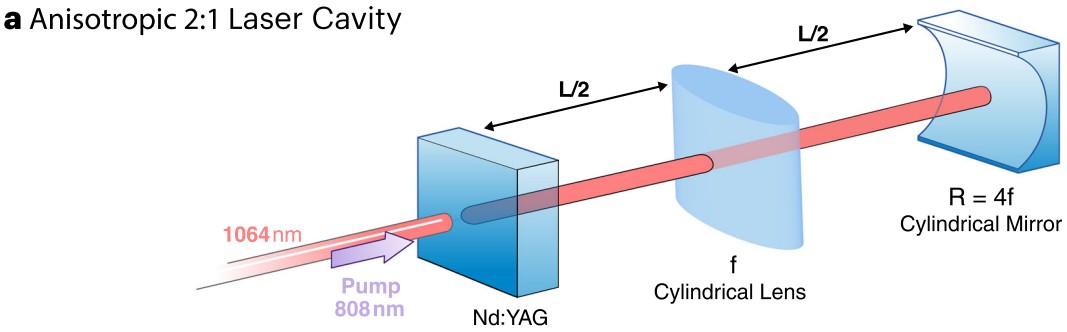

**b** Experimental Boyer-Wolf Gaussian Modes

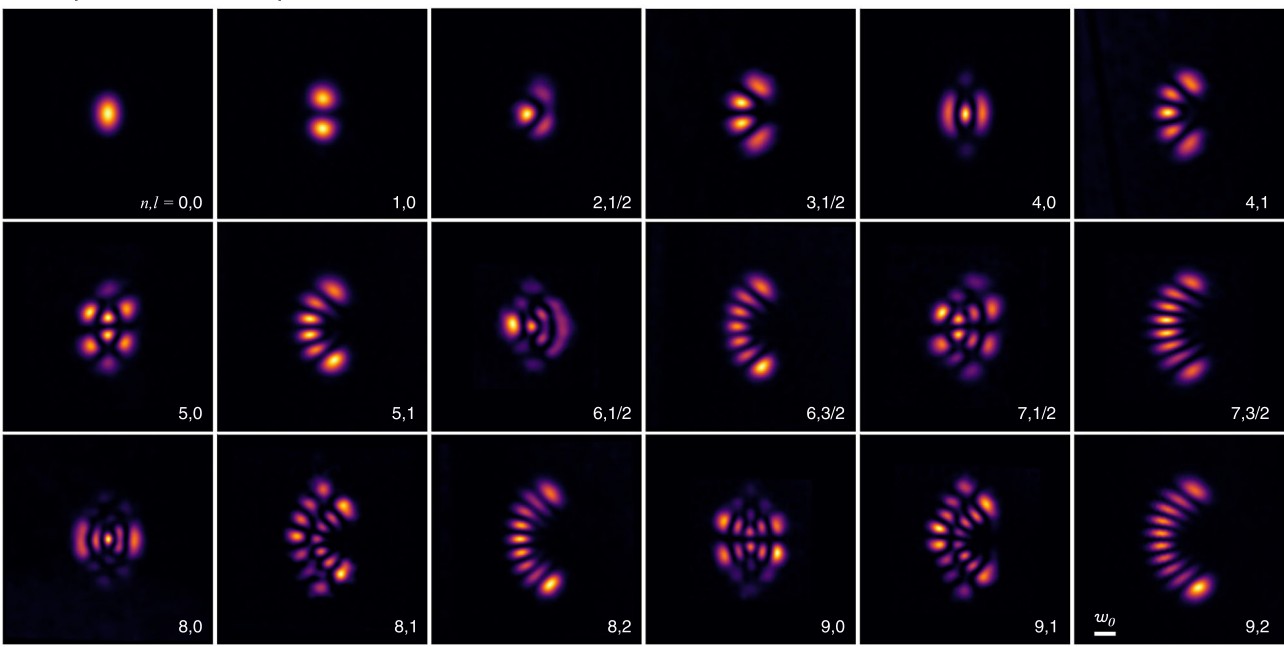

**Fig. 3 | Observation of Boyer-Wolf Gaussian modes. a** Experimental setup of the 2:1 anisotropic resonator cavity that lases in Boyer-Wolf Gaussian modes. The laser cavity of length $L$ is composed of a Nd:YAG flat crystal that acts as an output coupler and gain medium, a vertical cylindrical lens of focal length $f$, in the middle of the cavity, and a horizontal cylindrical mirror with $R = 2(2f)$. The cavity is pumped by focusing a 808 $nm$ light from a laser diode into the Nd:YAG crystal. **b** Measured laser beam intensity patterns of the first eighteen Boyer-Wolf Gaussian modes with $l > 0$, at the output coupler of the resonator. The mode order and degree $(n,l)$ are displayed in the bottom right corner of each panel. The scale of the modes is normalized to their width $w_0$ (inset of the bottom-right panel). The specific resonator length and width for each mode are provided in the Supplementary Information.

refractive index of the medium, and its eigenmodes are

$$
\mathrm{BWG}_{nl}^{aGRIN}(u,v,z) = c_{nl} \left( \frac{\sqrt{2}uv}{w_G} \right)^n \exp\left( -\frac{u^4 + v^4}{2w_G^2} \right) P_{nl}\left( \left[ 4\frac{u^2}{w_G} \right]^{-1} \right) P_{nl}
$$
$$
\left( -\left[ 4\frac{v^2}{w_G} \right]^{-1} \right) e^{-i\beta_n z},
$$

(4)

where $w_G^2 = 2/kn_0 a$, $\beta_n = a(2n+3)/2$, and $k = 2\pi/\lambda$.

The propagation of the Boyer-Wolf modes in an anisotropic GRIN medium is simulated in Fig. 4b. There, we can clearly see that the modes propagate without any diffraction within such media. Interestingly, these modes also present self-healing properties similar to structured beams in lens like media[54], see Supplementary Note 7.

### Transition to Weber parabolic beams

Finally, similar to how the LGM and IGM can transform into the Bessel and Mathieu nondiffractive beams[22,47], we found that the Boyer-Wolf Gaussian modes can transform into the Weber parabolic non-diffractive beams[21]. This metamorphosis occurs when the order, $n$, and

the waist size, $w_0$, tend towards infinity, while maintaining the ratio $k_t = 2\sqrt{n}/w_0$ constant. This ratio becomes the transverse wavevector of the nondiffractive beam, $k_t$. In Fig. 4c, we show this transition, there we can clearly see how, as the order and waist size of the Boyer-Wolf mode increase, it increasingly resembles a Weber parabolic non-diffractive beam. This transition enables the generation of high-power quasi-nondiffractive beams with parabolic geometry right at the laser source.

### Discussion

In summary, we have experimentally observed Boyer-Wolf Gaussian modes in a laser resonator. We designed a new class of laser resonator equivalent to a quantum two-dimensional 2:1 anisotropic harmonic oscillator by studying the isomorphism between stable laser cavities and quadratic Hamiltonians. The generated Boyer-Wolf Gaussian modes exhibit an inherent parabolic geometry that breaks the symmetry around the $y$-axis that is present in the HGM, LGM, and IGM, and creates a dark parabolic region around the $x$-axis. The observed Boyer-Wolf Gaussian modes not only represent a new fundamental family of modes of a laser resonator, but also the first realization of the eigen-solution of the 2D 2:1 anisotropic quantum harmonic oscillator in any

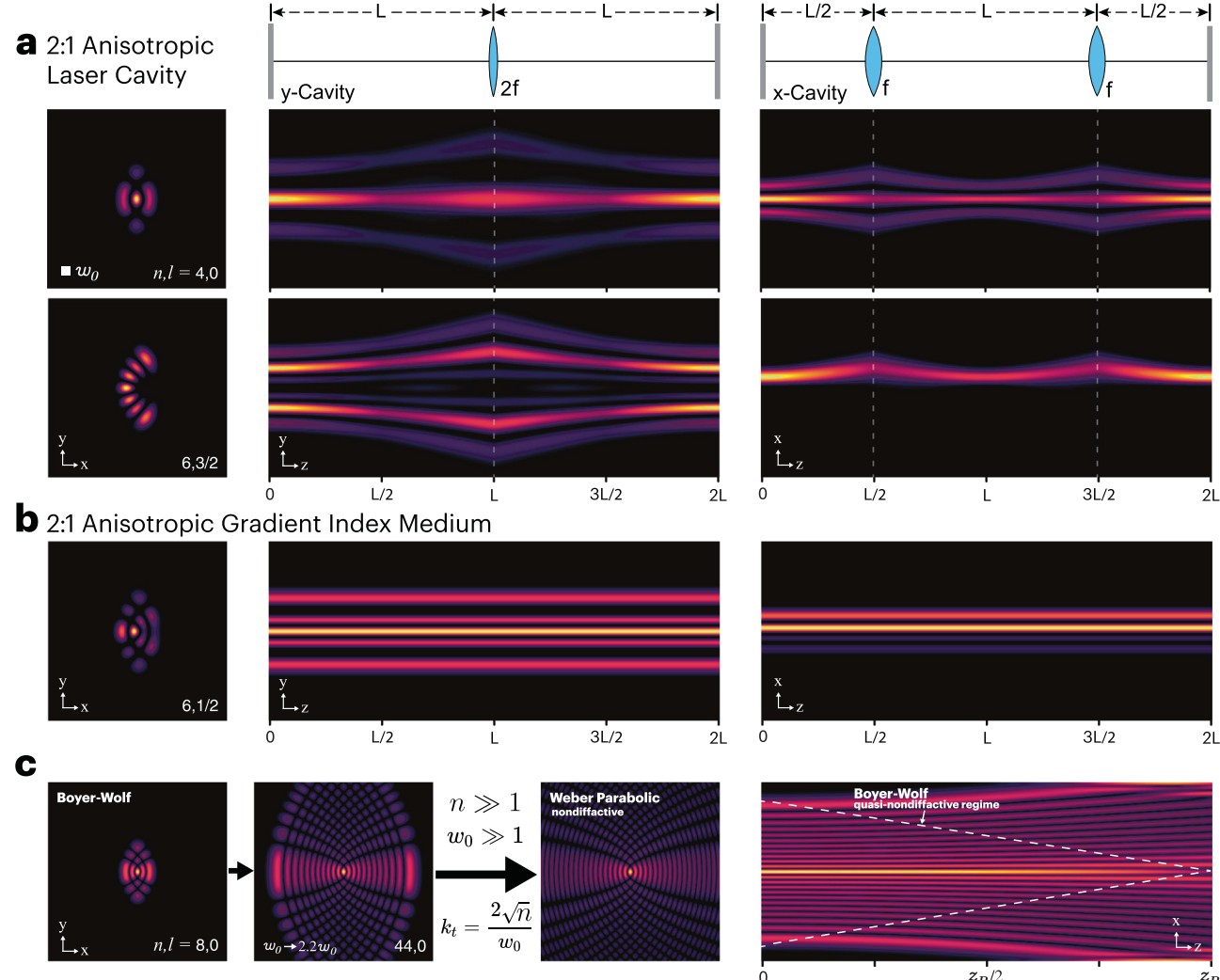

**Fig. 4 | Simulated propagation of Boyer-Wolf Gaussian modes in the 2:1 anisotropic laser cavity and in a 2:1 anisotropic GRIN medium. a** Left, amplitude of the transverse mode of the laser cavity at the output coupler. Right, $x$ and $y$-cross sections of a full round-trip propagation of the lasing mode inside the resonator cavity. The $y$-cylindrical mirror and the output flat mirror create a single cavity in the $y$-axis. While the $x$-cylindrical lens at the middle of the resonator creates two cavities, one with the flat mirror and one with the $y$-cylindrical mirror, as one can clearly observe from the propagation cross-sections. **b** Propagation of a Boyer-Wolf Gaussian mode in an anisotropic 2:1 GRIN medium. The Boyer-Wolf Gaussian modes are eigenmodes of such a GRIN medium and therefore propagate without diffraction. **c** Transmutation of the Boyer-Wolf Gaussian beams to a Weber parabolic nondiffracting beams. The metamorphosis takes place when the order, $n$, and the waist size, $w_0$, tend towards infinity, while maintaining the ratio $k_t = 2n/w_0$ constant, which becomes the transverse wavevector of the nondiffractive beam.

physical system and demonstrate experimentally the mathematical integrability of such system. We show that the Boyer-Wolf Gaussian modes are also eigenmodes of a 2:1 anisotropic gradient index medium, and therefore they must be present in any physical system—such as ultracold atoms, mechanical and acoustic systems—with a 2:1 anisotropic quadratic potential. We also find a transition that connects the Boyer-Wolf Gaussian modes to the Weber nondiffractive parabolic beams[21], allowing the generation of quasi-nondiffractive beams with parabolic geometry right at the laser source. The new Boyer-Wolf Gaussian modes serve as a fundamental element of structured light and open the door to new applications in laser micromachining, particle micromanipulation and optical communications.

## Methods

We generate the Boyer-Wolf Gaussian modes by building the laser resonator as depicted in Fig. 3a. The output coupler and gain medium is a 1.2% doped Nd:YAG crystal rod with dimensions of 5 mm length by 5 mm diameter, with a high-reflection coating at emission wavelength 1064 nm and an antireflection coating for the pump wavelength on one facet and an antireflection coating for both wavelengths on the second facet. To cool the Nd:YAG crystal, we wrap its perimeter with indium and place it inside a copper mounting attached to a water-cooling system at $16\,^{\circ}\mathcal{C}$. The crystal was end-pumped with a nLight Model Element e09i wavelength stabilized diode laser operating at 808 nm, which was collimated and focused into the Nd:YAG crystal, from outside the cavity. The incident pump power ranges from 0.5 to 3.5 W, and the output power is of the order of 10 mW, see Supplementary Note 8. The other mirror of the cavity is a $y$-axis silver-coated concave cylindrical mirror Ø1" with $f = 200\,mm$. At the middle of the cavity, we place a plano-convex cylindrical lens Ø1" with $f = 100\,mm$. We experimented with different lengths for the resonator, specifically, $L = 12, 15, 17, 20, 30\,cm$. To excite different BWG modes, we systematically changed the tilt angles of the cylindrical mirror and lens, see the Supplementary Notes 9, 10 for more details. We image the modes at the output coupler, by using a dichroic mirror ($R > 99.5\%$@1064 nm $T > 95\%$@808 nm @45°), a single lens, and a BladeCam-XHR CMOS (DataRay Inc.) camera ($\lambda = 355$ to 1150 nm, 6.5 × 4.9 mm, 3.2 μm pixels). We numerically filter the image noise by removing any spectral

frequency smaller than one percent of the peak spectral amplitude. A silicon wafer to cut off wavelengths below 1 $\mu m$ was used to suppress the remaining pump light.

## Data availability

All data that support the findings of this study are available within the paper and the Supplementary Information and are available from the corresponding author upon request.

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

## Acknowledgements

This material is based upon work supported by the National Science Foundation under Grant No. (2207964) M.A.B, K.T. M.A.B. acknowledges the profound impact of Kurt Bernardo Wolf's groundbreaking work in the field of group theory and symmetries in quantum mechanics and optics, which laid the foundation for our research. His enduring influence continues to inspire and guide us.

## Author contributions

M.A.B., K.T., and I.D. conceived the project idea. D.G. carried out the experiments and interpreted the results under the guidance and direction of K.T., O.M., I.D., and M.A.B. K.T. and M.A.B. performed the simulations, analysis, and compiled results. K.T. and M.A.B. wrote the manuscript and supplementary document in close consultation with D.G., O.M., and I.D.

## Competing interests

The authors declare no competing interests.
