## [Peer Review File · Nature Communications]

Observation of Boyer-Wolf Gaussian ModesREVIEWER COMMENTS

Reviewer #1 (Remarks to the Author):

In this manuscript four significant findings are presented:

- The experimental identification of a new class of fundamental laser modes in stable resonators: the Boyer-Wolf Gaussian modes.
- The design of a laser resonator analogous to a two-dimensional quantum anisotropic harmonic oscillator, with a frequency ratio of 2:1.
- It is demonstrated that Boyer-Wolf Gaussian modes also function as eigenmodes within a medium with an anisotropic refractive index gradient, exhibiting a 2:1 ratio between the parabolic index of refraction profiles along each axis.
- A transition is identified that connects Boyer-Wolf Gaussian modes with non-diffractive Weber parabolic beams, highlighting the fundamental role of Boyer-Wolf modes in structured light.

To the best of my knowledge, Tschernig et al. present the Boyer-Wolf Gaussian modes for the first time, obtained from the corresponding time-independent Schrödinger equation of the quantum two-dimensional anisotropic harmonic oscillator with a 2:1 frequency ratio in parabolic coordinates. Subsequently, the authors elegantly design a laser resonator to generate these modes and demonstrate that experimental results align well with the theoretical predictions. They provide an analytical model to rigorously illustrate the effect and validate it using a straightforward optical setup. Furthermore, the manuscript demonstrates that the Boyer-Wolf Gaussian modes function as eigenmodes within an anisotropic gradient refractive index medium, akin to fundamental Hermite-Gauss, Laguerre-Gauss, and Ince-Gauss Modes. Lastly, the authors show that Boyer-Wolf modes are foundational elements of structured light. The Supplementary Information document contains an exhaustive analysis of the Boyer-Wolf Gaussian modes, including resonator propagation, gradient refractive index propagation, and free-space propagation of these modes.

I find the demonstrations both interesting and highly relevant to the optical community. This work is likely to be a cornerstone, sparking a series of novel ideas in this specific field. Therefore, I believe the manuscript is well-suited for the Nature Communications audience. Nevertheless, I have listed below some minor suggestions for the authors to consider in a revised manuscript.

1. Before Eq. (2), the authors state: "The solution to Eq. (1) in parabolic coordinates (u,v) , $x=(u^2+v^2)$, $y=uv...$ ". However, in Ref [44] and in the Supplementary Information of this manuscript, it is noted that the correct expression is $x = (u^2-v^2)$. Please correct this error accordingly.

2. While the manuscript thoroughly discusses the advantages of the employed resonator design and experimental setup, it is crucial to also address potential limitations and challenges for a more comprehensive evaluation. This disadvantages, that are not mentioned in this version of the manuscript, will provide a balanced perspective for readers and contribute to a more thorough understanding of the methodology.

3. In a recent publication (<https://doi.org/10.1364/OE.507030>), the self-healing capability of structured beams, such as higher-order Gaussian modes, Laguerre-Gaussian beams, and Hermite-Gaussian beams, has been investigated when partially obstructed at the onset of propagation in a lens-like medium with obstacles of various shapes. It would be highly beneficial to incorporate Boyer-Wolf Gaussian modes into this context and provide an explanation regarding their potential for self-healing and the mechanisms involved. I am confident that the authors can address this question seamlessly, offering additional insights into whether Boyer-Wolf Gaussian modes exhibit self-healing characteristics and elucidating the underlying processes. This expansion of the discussion would significantly contribute to a more comprehensive understanding of self-healing phenomena in

structured beams.

Overall, I find the presentation of this manuscript to be quite good. Nevertheless, there is room for improvement, and making these minor corrections would enhance the overall quality of the manuscript, resulting in a more robust and comprehensive final product.

Reviewer #2 (Remarks to the Author):

This paper is experimentally obtained a new stable laser mode in the resonator - the Boyer-Wolf Gaussian mode, and the experimental results were in good agreement with the theoretical predictions. Moreover, the pattern of Boyer-Wolf Gaussian mode can successfully transform to Weber nondiffractive parabolic beams, indicating that this new type of beam is a basic element of structured light. Although this article has these innovative elements, the content of the paper is too thin, and the discussion of the experiments and methods is not detailed enough. It does not meet the criteria of Nature Communications. I think this work would be more suitable for publication in more specialized journals, such as Opt. Express or Opt. Letters. My comments to the paper are listed below:

- 1, The authors only select some typical length (i.e., $L=12,15,17,20$ and 30 cm) of the resonator. However, the modes will be different when the cavity changes. Whether the results in Fig. 3b are obtained when the cavity length is fixed? If so, the results with other cavity lengths should be discussed.
- 2, The article mentioned that the advantage of the cavity design in Fig.3a is that for any pair of cylindrical lenses with $f_y=2f_x$, the resonant cavity is stable for any $L\leq 4f_x$. Please give a specific explanation.
- 3, The different sizes of mode-to-pump ratio on the Nd:YAG laser crystal will significantly affect the quality of the output beam, so, what is the optimal pump spot size in this laser device? What is the impact on the output results of this new Gaussian mode when using other pump spot sizes?
- 4, The results in Fig. 4 are from numerical simulation or experimental observation? I think the authors should clearly explain how they obtain such results.
- 5, I am amazing that the numerical results in Fig. 2 and the experimental results in Fig. 3b can matched with each other quite well. I think the author should provide a more detail about their experimental setup, and the size of pattern in Fig. 3b should be provided.

Reviewer #3 (Remarks to the Author):

The authors found an interesting new set of structured separable cavity modes for a cavity with a ratio of 2 to 1, based on mathematical work from 1975 by Boyer and Wolf. They implemented an anisotropic laser cavity that supports the desired modes, which agree impressively well with theory. The main contribution of this work is clearly presented.

I do have some comments on some aspects that are not central to the work, and some observations that are more related to the new contribution. I begin with two points that apply to the introductory comments and not to the main contribution.

1) There is one point that the authors mention repeatedly and that in my opinion is not strictly correct as written, or at least it is not sufficiently well justified. The statement in question is that there are only "three different fundamental families of transverse modes" of the usual rotationally-symmetric cavities: HGM, LGM and IGM. These families of modes are certainly important and have served as the basis of large amounts of work, but given the degeneracy of the system (clearly indicated by the authors), they are in principle not more fundamental than others.

In my opinion, the correct statement is that these are the only three families of transverse modes that can be written as separable functions. This is a mathematical statement rather than a physical one.

From a physical point of view, one aspect that makes these three families a bit more special is that they are naturally selected when the cavity presents small amounts of simple typical aberrations like astigmatism (which leads to HG), spherical aberration (which selects LGM) or a combination of both (which leads to IG) as was shown in [1]. (Notice that, interestingly, spherical aberration is a higher order aberration and cannot hence be described with the ABCD formalism, and yet a perturbative treatment does lead to the LGM or IGM.) However, other more complex aberrations might lead to other modes that, given the unitarity of the system, this will result in complete orthonormal bases. For example, a combination of a small amount of astigmatism and a small rotation (e.g. in a misaligned ring cavity) can select what is known as Hermite-Laguerre-Gauss modes, which also include HGM and LGM as special cases and are then a family of complete bases with closed-form expressions [3,4], although these expressions are not separable mathematically.

Again, I do not mean to downplay the importance of the three families mentioned (which in any case are not the central contribution of this work). However, I fear that a reader that is new on the subject might understand the statements as currently written as meaning that these are the only three types of modes that can be produced by the cavity, which is not the case. Note that there is indeed a sentence in the manuscript that reads "For this reason, spherical optical cavities only support three families of fundamental modes, the Hermite, Laguerre and Ince-Gaussian modes".

2) Similarly, while I greatly appreciate the importance of Gaussian modes, I think that the statement that "Gaussian modes are the foundation of structured light" a bit too strong. It is true that many other standard solutions, such as accelerating or propagation-invariant (sometimes referred to incorrectly as "diffraction-free") beams can be thought of as limiting cases of these modes [5]. There are many aspects of structured light that do not correspond to this limit. In general, all light is structured! Again, this is surely a minor point but clarity is important, and I think one can make the case for the great importance of Gaussian modes without making such a strong claim.

Let me now make some comments related to the main contribution, which is the BWM. From what I have checked, the results seem to all be correct. Nevertheless, I would like to make a couple of suggestions that the authors could consider here and/or in subsequent work.

3) This comment echoes comment 1. From the current version, a reader might be under the impression that, once the cavity is proposed and appropriately manipulated, the desired modes emerge. However, as stressed in the article by Boyer and Wolf, this system presents "accidental degeneracy", so like the standard isotropic cavity its modes can be decomposed in an infinite number of bases. One basis that immediately comes to mind is that of the (anisotropic) HGM aligned with the directions of the anisotropy. This basis is not mentioned in the main body, but only in the Supplementary Information. These modes are likely naturally selected if the strength of the oscillator is detuned from the ideal 2:1. In the main manuscript the authors mention that they selected higher order BWM by slightly misaligning the cavity "in different ways". I would guess that this took a significant amount of trial and error. I would encourage the authors to think of what perturbation of the system maps onto the operator that defines the modes and that commutes with the anisotropic harmonic oscillator Hamiltonian, in a way similar to what was done in [1] for the IGM.

4) While this last issue is discussed briefly in the Supplementary Information, I think it is important to mention it more clearly on the main manuscript as well.

There is an important difference on the behavior of the BWM and the modes for the standard rotationally symmetric cavities (or GRIN waveguides), such as HGM, LGM and IGM. For the latter, the modes of the cavity or the GRIN medium are also modes in a generalized sense of free space, because free paraxial propagation can be mapped onto an isotropic 2D harmonic oscillator. This is the reason why standard Gaussian modes (be them HG, LG, IG, HLG or others) maintain their intensity profile up to an expansion and corresponding decrease in intensity. Amongst other things, this means that these modes have Fourier transforms with the same shape. The new modes do not present strictly the property of self similarity under free propagation, so the profiles plotted correspond to the profile at some plane within the cavity. As it is stated in the Supplementary Information, these modes can be

expressed in terms of anisotropic HGM, which accumulate different Gouy phases. Interestingly, Fig. 14 of the Supplementary Document shows that the far-field distribution (and hence the Fourier transform) is a non-uniformly scaled version of the original mode. This property is trivial to show for the HGM in Eq. (45), and I believe that the derivation in Eqs. (43-57) shows it, but does not spell it out clearly. That is, going from the field to its Fourier transform via a Fresnel or Fractional Fourier transform, the modes return to their same shape up to a scaling, but in between they likely take different shapes. This is hard to judge from the longitudinal plots in Fig. 14, where it is difficult to tell if the relative importance of the intensity maxima changes with z . If the modes do preserve their shape up to an anisotropic stretching, this is worth mentioning, as they would be interesting as free-space modes, not only as cavity modes. Again, even a brief mention of this behavior in the main manuscript (with more detail in the Supplementary Information) would be useful.

References of potential interest to the authors. In particular the first two are directly relevant to the subject.

- [1] R. Gutiérrez-Cuevas et al., Phys. Rev.A 107, L031502 (2023)
- [2] M. Jaffe et al., Phys. Rev.A 104, 013524 (2021).
- [3] E. G. Abramochkin and V. G. Volostnikov, J. Opt. A: Pure Appl. Opt. 6, S157 (2004).
- [4] R. Gutiérrez-Cuevas et al., J. Opt. 21, 084001 (2019).
- [5] M. A. Alonso and M. R. Dennis, Optica 4, 476 (2017).

Observation of Boyer-Wolf Gaussian Modes

Response to Comments made by Referees

We thank all three referees for their favorable reviews and their insightful comments.

Thank you all for your positive feedback.

We generally agree with the comments made by the referees and we revised the manuscript according to the suggestions, which have indeed improved our manuscript considerably. In the following sections, we address each of the comments individually and explain the changes made in the manuscript and in the Supplementary Information in response to each comment.

Point by Point Response to the Comments of the Referees

We now address each comment individually. For each point, we first present the referee's original comment, followed by our response. To facilitate this process, we have assigned numbers to every individual point raised by each referee, organized in the order of their appearance in the report of each referee.

REVIEWER COMMENTS

Reviewer #1 (Remarks to the Author):

In this manuscript four significant findings are presented:

- The experimental identification of a new class of fundamental laser modes in stable resonators: the Boyer-Wolf Gaussian modes.
- The design of a laser resonator analogous to a two-dimensional quantum anisotropic harmonic oscillator, with a frequency ratio of 2:1.

- It is demonstrated that Boyer-Wolf Gaussian modes also function as eigenmodes within a medium with an anisotropic refractive index gradient, exhibiting a 2:1 ratio between the parabolic index of refraction profiles along each axis.
- A transition is identified that connects Boyer-Wolf Gaussian modes with non-diffractive Weber parabolic beams, highlighting the fundamental role of Boyer-Wolf modes in structured light.

To the best of my knowledge, Tschernig et al. present the Boyer-Wolf Gaussian modes for the first time, obtained from the corresponding time-independent Schrödinger equation of the quantum two-dimensional anisotropic harmonic oscillator with a 2:1 frequency ratio in parabolic coordinates. **Subsequently, the authors elegantly design a laser resonator to generate these modes and demonstrate that experimental results align well with the theoretical predictions.** They provide an analytical model to rigorously illustrate the effect and validate it using a straightforward optical setup. Furthermore, the manuscript demonstrates that the Boyer-Wolf Gaussian modes function as eigenmodes within an anisotropic gradient refractive index medium, akin to fundamental Hermite-Gauss, Laguerre-Gauss, and Ince-Gauss Modes. Lastly, the authors show that Boyer-Wolf modes are foundational elements of structured light. **The Supplementary Information document contains an exhaustive analysis of the Boyer-Wolf Gaussian modes,** including resonator propagation, gradient refractive index propagation, and free-space propagation of these modes.

I find the demonstrations both interesting and highly relevant to the optical community. This work is likely to be a cornerstone, sparking a series of novel ideas in this specific field. Therefore, I believe the manuscript **is well-suited for the Nature Communications audience.**

Our response:

We thank the referee for the positive reaction on our manuscript. We are pleased that the referee found our work both interesting, highly relevant to the optical community, and well-suited for publication in Nature Communications. As described below, we have carefully implemented the referee's suggestions, which have significantly enhanced the quality of our work.

Nevertheless, I have listed below some minor suggestions for the authors to consider in a revised manuscript.

1. Before Eq. (2), the authors state: "The solution to Eq. (1) in parabolic coordinates (u,v) , $x=(u^2+v^2)$, $y=uv\dots$ ". However, in Ref [44] and in the Supplementary Information of this manuscript, it is noted that the correct expression is $x = (u^2-v^2)$. Please correct this error accordingly.

Our response:

We thank the referee for carefully reviewing our manuscript and for finding this typo. We have corrected the sign in the equation accordingly.

2. While the manuscript thoroughly discusses the advantages of the employed resonator design and experimental setup, it is crucial to also address potential limitations and challenges for a more comprehensive evaluation. This disadvantages, that are not mentioned in this version of the manuscript, will provide a balanced perspective for readers and contribute to a more thorough understanding of the methodology.

Our response:

We thank the referee for bringing this matter to our attention. Similar to a spherical resonator, the primary limitation here is the presence of high-order aberrations. As we describe in the manuscript, the theoretical description of any paraxial ABCD resonator only considers quadratic terms in position and momentum. Consequently, this implies that higher-order aberrations are missing in the conventional ABCD ray formulation. The overall effect in our work, as in any other spherical laser resonator, is that the misalignments that we introduce to generate higher order modes need to remain within the paraxial regime, i.e., small misalignment angles. We have now included a paragraph to the manuscript where we discuss the limitation of paraxial resonator design and experimental setups. Additionally, we highlight approaches for studying this system beyond the paraxial regime.

Paragraph:

“The primary limitation of this formalism is that the theoretical description of ABCD systems only considers “quadratic optics” and does not include higher-order aberrations, such as mirror imperfections and non-paraxial corrections. Here, and generally, the paraxial approximation is entirely adequate because deviations from this approximation are usually small when the resonator mode waist w_0 is significantly larger than the wavelength λ . Therefore, for our paraxial resonator, these corrections only become important for misalignments outside the paraxial regime, for very high-order modes, and for resonator configurations with very small waist, i.e. $L \sim 2f_y$. A complete theory of resonator aberrations for nonparaxial beam propagation and optical elements beyond-quadratic mirrors and lenses can be found in Ref. [48].”

3. In a recent publication (<https://doi.org/10.1364/OE.507030>), the self-healing capability of structured beams, such as higher-order Gaussian modes, Laguerre-Gaussian beams, and Hermite-Gaussian beams, has been investigated when partially obstructed at the onset of propagation in a lens-like medium with obstacles of various shapes. It would be highly beneficial to incorporate Boyer-Wolf Gaussian modes into this context and provide an explanation regarding their potential for self-healing and the mechanisms involved. I am confident that the authors can address this question seamlessly, offering additional insights into whether Boyer-Wolf Gaussian modes exhibit self-healing characteristics and elucidating the underlying processes. This expansion of the discussion would significantly contribute to a more comprehensive understanding of self-healing phenomena in structured beams.

Our response:

We thank the referee for bringing to our attention this interesting recent work. Following this reference, we have conducted a similar analysis of the self-healing properties of the high-order Boyer-Wolf Gaussian modes propagation in a 2:1 GRIN medium. As the referee suggested, we found that similar to the higher-order Laguerre, Hermite, or Ince-Gaussian beams, the Boyer-Wolf modes also exhibit self-healing properties. The mechanism is the same as the one presented in the reference above. The main difference is that since the BW modes are eigenmodes of a 2:1 GRIN medium, as opposed to a 1:1

GRIN medium, the self-healing process is dictated by the axis with the larger period of oscillations.

We have added a sentence to the manuscript addressing the self-healing properties of the BW based on this new reference. More importantly, **we have included an entirely new section in the Supplementary Material (Section 7.1)**, where we conduct a more thorough analysis of the self-healing properties of the BW modes following the suggested reference. Furthermore, we illustrate this phenomenon with **two new supplementary figures** (Supp. Fig. 19 and 20).

Sup. Figure 20 | Partial and fully self-healing of higher order BWG modes. We show several x - y -slices of the propagation shown in Fig. (1). At $z = L_z/4$ and $z = 3L_z/4$ we observe (almost) complete healing of the beam profile after encountering the circular block. At other distances we observe only partial healing or, in the extreme case, the recurrence of the initial circular block.

Sup. Figure 19 | Propagation of partially blocked high-order BWG mode. We show the x - z -slice of the propagation of the $n = 44$, $l = 0$ BWG mode in a 2:1 GRIN medium with $a = 1.1272$ 1/m and $\lambda = 1064$ nm. The blocked region causes the emergence of shadows that propagate along sinusoidal paths and reconstitute the initial blocked circle at $z = L_z = 2.786$ m.

Overall, **I find the presentation of this manuscript to be quite good**. Nevertheless, there is room for improvement, and making **these minor corrections** would **enhance the overall quality of the manuscript, resulting in a more robust and comprehensive final product**.

Our response:

We are glad that the referee found our manuscript quite good. Indeed, implementing the referee's suggestion has enhanced the quality of our manuscript.

Reviewer #2 (Remarks to the Author):

This paper is experimentally obtained a new stable laser mode in the resonator - the Boyer-Wolf Gaussian mode, and the experimental results were in good agreement with the theoretical predictions. Moreover, the pattern of Boyer-Wolf Gaussian mode can successfully transform to Weber nondiffractive parabolic beams, indicating that this new type of beam is a basic element of structured light. Although this article has these innovative elements, the content of the paper is too thin, and the discussion of the experiments and methods is not detailed enough. It does not meet the criteria of Nature Communications. I think this work would be more suitable for publication in more specialized journals, such as Opt. Express or Opt. Letters.

Our response:

We are confused by the comments of the referee. We believe the referee probably did not realize that there was a supplementary section to the paper. As the reviewer #1 said "the Supplementary Information document contains an exhaustive analysis of the Boyer-Wolf Gaussian modes, including resonator propagation, gradient refractive index propagation, and free-space propagation of these modes." Our supplementary material is very complete and discusses in detail the theoretical and experimental methods that we used to properly design the resonator. Some of the concerns raised by the referee are addressed in this supplementary material. For instance, in the supplementary material we have a detailed study of the stability conditions of our resonator. We find the other comments of the referee very helpful to improve the presentation and clarity of our work. As described below, we have carefully addressed each of the comments.

My comments to the paper are listed below:

1, The authors only select some typical length (i.e., $L=12,15,17,20$ and 30 cm) of the resonator. However, the modes will be different when the cavity changes. Whether the results in Fig. 3b are obtained when the cavity length is fixed? If so, the results with other cavity lengths should be discussed.

Our response:

We thank the reviewer for this comment, which helped us to improve the clarity of our results. As the reviewers point out, it was not clear in the manuscript what length of the resonator we were using in Fig. 3. To excite different modes, we use different resonator lengths. The idea is that to excite different modes, one can control the pump size, since different modes have lobes of different sizes. Alternatively, one can keep the size of the pump constant and change the size of the resonator. This effectively changes the size of the modes and, consequently, the relative size between the pump and the modes. We found this approach more stable and easier to implement experimentally than changing the pump size. Changing the pump size requires addressing issues such as saturation, especially if the pump size becomes very small but maintains the same total power.

Of course, by changing the length of the resonator, the width of the modes also changes. This variation follows the formula provided in the manuscript, $w_0^2 = 2k^{-1}\sqrt{L(2f_y - L)}$, which was derived in the supplementary material and illustrated in Supplementary Fig. 9. In Fig. 3, we normalize the scale of the transverse plane to the width of each mode to allow a better visualization of all the modes at the same scale.

We have clarified in the manuscript that the modes are normalized to their own width, and that the modes correspond to different sets of resonator lengths. We have added a **new Section 9 in the Supplementary Material** where we describe in more detail how we generate the high-order modes. We include a table where we provide the specific length of the resonator and the width for each of the modes. Notice that the widths are very similar across all modes.

2, The article mentioned that the advantage of the cavity design in Fig.3a is that for any pair of cylindrical lenses with $f_y=2f_x$, the resonant cavity is stable for any $L \leq 4f_x$. Please give a specific explanation.

Our response:

The explanation regarding the stability of the resonator was present in Section 4.1 of the supplementary material. There, we provided a concise derivation of the stability based on the known stability results of Fabry Perot resonators. The stability condition was depicted in Sup. Fig. 8. **We have expanded Section 4.1 of the Supplementary Material** to provide a more comprehensive understanding of the stability conditions of our resonator.

3, The different sizes of mode-to-pump ratio on the Nd:YAG laser crystal will significantly affect the quality of the output beam, so, what is the optimal pump spot size in this laser device? What is the impact on the output results of this new Gaussian mode when using other pump spot sizes?

Our response:

Yes, this is an important point. This comment relates to comment #1 above. The effect of reducing the pump size is that different sizes will excite different modes. Let us elaborate: for example, in Fig. 3 we can clearly see that the strongest lobe of each mode varies in size. Therefore, the mode whose intensity profile overlaps most with the pump profile will have an advantage in lasing. As we mention above, instead of controlling the pump size to select the lasing mode, we change the size of the cavity. This effectively changes the width of the mode and alters the scale between the pump size and the mode width. We have elaborated on the effect of the pump in the **new Section 9 of the supplementary material.**

4, The results in Fig. 4 are from numerical simulation or experimental observation? I think the authors should clearly explain how they obtain such results.

Our response:

We thank the reviewer for highlighting this point, which could have caused confusion. The results in Fig. 4 are from simulations. In the manuscript, each time we refer to Fig. 4,

we have pointed out that these are simulations. However, to avoid any misunderstanding, as the reviewer suggested, we have changed the title of Fig. 4 to “**Simulated** Propagation of Boyer-Wolf Gaussian Modes in the 2:1 ...”

5, I am amazing that the numerical results in Fig. 2 and the experimental results in Fig. 3b can matched with each other quite well. I think the author should provide a more detail about their experimental setup, and the size of pattern in Fig. 3b should be provided.

Our response:

We are glad that the reviewer found our experimental results to match the theoretical ones quite well. In a **new Section 9 of the supplementary material**, we have included additional information about the experimental setup and the methods used to excite the BW modes. There, we have also added a table detailing the corresponding resonator length for each lasing mode and its respective width.

Reviewer #3 (Remarks to the Author):

The authors **found an interesting new set of structured separable cavity modes** for a cavity with a ratio of 2 to 1, based on mathematical work from 1975 by Boyer and Wolf. They implemented an anisotropic laser cavity that supports the desired modes, **which agree impressively well with theory**. The main contribution of this work is clearly presented.

Our response:

We thank the referee for the positive reaction to our manuscript. We are pleased that the referee found our work interesting and that the experimental results agree impressively well with the theory. We thank the referee for all his interesting and deep comments. As described below, we have carefully implemented the referee's suggestions, which have helped us to improve the quality of our work.

I do have some comments on some aspects that are not central to the work, and some observations that are more related to the new contribution. I begin with two points that apply to the introductory comments and not to the main contribution.

1) There is one point that the authors mention repeatedly and that in my opinion is not strictly correct as written, or at least it is not sufficiently well justified. The statement in question is that there are only “three different fundamental families of transverse modes” of the usual rotationally-symmetric cavities: HGM, LGM and IGM. These families of modes are certainly important and have served as the basis of large amounts of work, but given the degeneracy of the system (clearly indicated by the authors), they are in principle not more fundamental than others. In my opinion, the correct statement is that these are the only three families of transverse modes that can be written as separable functions. This is a mathematical statement rather than a physical one. From a physical point of view, one aspect that makes these three families a bit more special is that they are naturally selected when the cavity presents small amounts of simple typical aberrations like astigmatism (which leads to HG), spherical aberration (which selects LGM) or a combination of both (which leads to IG) as was shown in [1]. (Notice that, interestingly, spherical aberration is a higher order aberration and cannot hence be described with the ABCD formalism, and yet a perturbative treatment does lead to the LGM or IGM.) However, other more complex aberrations might lead to other modes that, given the unitarity of the system, this will result in complete orthonormal bases. For example, a combination of a small amount of astigmatism and a small rotation (e.g. in a misaligned ring cavity) can select what is known as Hermite-Laguerre-Gauss modes, which also include HGM and LGM as special cases and are then a family of complete bases with closed-form expressions [3,4], although these expressions are not separable mathematically.

Again, I do not mean to downplay the importance of the three families mentioned (which in any case are not the central contribution of this work). However, I fear that a reader that is new on the subject might understand the statements as currently written as meaning that these are the only three types of modes that can be produced by the cavity, which is not the case. Note that there is

indeed a sentence in the manuscript that reads "For this reason, spherical optical cavities only support three families of fundamental modes, the Hermite, Laguerre and Ince-Gaussian modes".

Our response:

We understand the reviewer's perspective and observations. We thank the referee for pointing out that this comment is about our interpretation and presentation of the "Gaussian beams," and not about the main contribution of the paper.

Please allow us to elaborate on our point of view. The fact that the three families of transverse modes can be written as separable functions is not only a mathematical statement but also a physical one. From all the work in integrable system it is known that a system is separable in two-dimensions implies that it has at least two conserved **physical quantities**. This means, that these solutions are eigenfunctions of two operators, one being the Hamiltonian and the other one a second order differential operator common to each family. For example, the most well know example is the orbital angular momentum of the Laguerre Gaussian modes. One can take a more physical equivalent approach and say that the fact that these families of modes have an extra conserved quantity implies that they can be separable and have an appealing mathematical structure. For this reason, even though one can create new families by rearranging the degenerate modes at each level, any of these families will have an extra conserved physical quantity. In any case, we have included the word "separable" when referring to this families to make the appropriate distinction. This is, as the referee suggests, for clarity, we have added the word 'separable' to the following sentence: ".spherical optical cavities only support three separable families..."

We are taking the definition of "**fundamental**" as "*basic principle or foundation, then something that is fundamental serves as a basic, underlying principle or foundation upon which other things are built or based*". We think that it is acceptable to say that these three families of modes are the fundamental ones. Since due to their physical symmetries they have a nice mathematical structure that allow us to use them as a building block for any other combination of states in the system. For example, the solutions for astigmatic resonators, references [3,4] that the referee mentions, are given as summations of Hermite Gaussian modes. This corroborates our definition of these modes as fundamental, since they are the foundational modes upon which other modes are built.

2) Similarly, while I greatly appreciate the importance of Gaussian modes, I think that the statement that “Gaussian modes are the foundation of structured light” is a bit too strong. It is true that many other standard solutions, such as accelerating or propagation-invariant (sometimes referred to incorrectly as “diffraction-free”) beams can be thought of as limiting cases of these modes [5]. There are many aspects of structured light that do not correspond to this limit. In general, all light is structured! Again, this is surely a minor point but clarity is important, and I think one can make the case for the great importance of Gaussian modes without making such a strong claim.

Our response:

We totally understand and appreciate the comment of the reviewer. Here we are going a little bit more to the philosophical regime. For example, it is true that all light is structured, but some light fields are way more structured than others. But as the referee suggests, to clarify the importance of the Gaussian beam without making such a strong claim we have changed the sentence to: “*Gaussian modes are a cornerstone of structured light.*” And “... *they serve as a cornerstone of structured light.*”

We hope the reviewer finds this rewording more appropriate.

Let me now make some comments related to the main contribution, which is the BWM. From what I have checked, the results seem to all be correct. Nevertheless, I would like to make a couple of suggestions that the authors could consider here and/or in subsequent work.

3) This comment echoes comment 1. From the current version, a reader might be under the impression that, once the cavity is proposed and appropriately manipulated, the desired modes emerge. However, as stressed in the article by Boyer and Wolf, this system presents “accidental degeneracy”, so like the standard isotropic cavity its modes can be decomposed in an infinite number of bases. One basis that immediately comes to mind is that of the (anisotropic) HGM aligned with the directions of the anisotropy. This basis is not mentioned in the main body, but only in the Supplementary Information. These modes are likely naturally selected if the strength of the oscillator is detuned from the ideal 2:1. In the main manuscript the authors mention that they selected higher order BWM by slightly misaligning the cavity “in different ways”. I would

guess that this took a significant amount trial and error. I would encourage the authors to think of what perturbation of the system maps onto the operator that defines the modes and that commutes with the anisotropic harmonic oscillator Hamiltonian, in a way similar to what was done in [1] for the IGM.

Our response:

We are thankful for this comment. It is very important to note that we are not modifying the 2:1 ratio of the cavity. Let us elaborate: since the misalignments of the resonator occur in the tipping and tilting of the mirrors and cylindrical lenses, these do not affect the ideal 2:1 ratio of our system, always maintaining the resonator close to accidental degeneracy. The tipping and tilting misalignments introduce linear terms in the Hamiltonian, but not quadratic ones. To explain this in detail, we have added a new section to the supplementary material (Section 10: Mode Selection via Cavity Misalignment) where we perform an extensive analysis of the impact of these misalignments. We study a Fabry-Pérot cavity with one misaligned mirror, using the extended ABCDEF formalism. After translation into the Hamiltonian picture, we find, as expected, that the potential appears spatially shifted, i.e. $k^2 \omega^2 (x-x_0)^2$, and the total energy (Gouy phase) shifts by an amount dependent on the displacement and tilt angle of the mirror. Additionally, the phase front of the beam acquires a linear tilt, also dependent on the misalignment parameters. The only way to break the 2:1 ratio would be to change the ratio of focal distances between our cylindrical lens and the cylindrical mirror or to move the cylindrical lens from the center of the cavity. We are not considering any of this as we aim to maintain the 2:1 anisotropic cavity. This is the main reason our resonator can lase in BWG modes.

Most importantly, the selection mechanism of the lasing mode is not only related to the cavity. As the reviewer suggests in Ref. [1], there are perturbations that can "lock" the cavity to a specific family of modes. However, another mechanism of selection is the pump profile and mode competition. In very simple terms, the mode that has more overlap with the pump will be the one that lases. The mode selection mechanism in our setup is rooted in the overlap of the pump spot with the selected mode. In other words, by displacing the pump spot, or equivalently, by displacing/tilting the optical axis (which we achieve by tilting the mirror and lens), we control the overlap of the pump spot with the modes. Then,

naturally, the mode with the highest overlap will win over many cycles inside the partially lossy cavity. Let us elaborate on this point. As we mentioned in our previous response, we have included new section 9 in the supplementary material, where we describe our (mis)alignment strategies.

Firstly, we change the cavity length to modify the size of the modes and, thereby, their overlap with the pump beam spot. For example, decreasing the cavity length reduces the mode size and thus improves the overlap of higher-order modes with the pump beam spot. Secondly, to understand how these misalignments affect the overlap between the pump and the individual modes, we performed a numerical calculation to determine which modes are most favored under different pump displacements. As shown in the new Figure 22 of the supplementary material (and below), the BWG modes are more favored than the anisotropic HG modes in certain regions of pump displacement. Specifically, we illustrate the difference in overlaps between the best BWG and the best HG mode up to $n=9$. Here, we used a Gaussian beam spot of 250 μm radius and a cavity length of 20 cm, closely

Sup. Figure 14 | Diagram of the parabolic momentum of the BWG modes. We show the eigenvalues, or parabolic momenta, of the BWG modes with respect to the parabolic symmetry operator. Here we use normalized units as in Ref. [3].

matching the parameters used in the actual experiment. While these results do not preclude the presence of higher-order perturbations that might naturally select the BWG modes, we believe that the pump overlap sufficiently explains the emergence of the BWG modes in our setup. This is also consistent with the fact that some trial and error in adjusting the mirror and lens tilt angle was necessary to achieve all the BWG modes up to $n=9$. Finally, we have added sentences in the methods section.

“To excite different BWG modes we systematically changed the tilt angles of the cylindrical mirror and lens, see supplementary material for details.”

and in the main body of the manuscript:

“This was achieved by adjusting the cavity length and shifting the position of the optical axis relative to the pump beam, through slight tilting and displacement of the cylindrical mirror and lens.”

4) While this last issue is discussed briefly in the Supplementary Information, I think it is important to mention it more clearly on the main manuscript as well.

There is an important difference on the behavior of the BWM and the modes for the standard rotationally symmetric cavities (or GRIN waveguides), such as HGM, LGM and IGM. For the latter, the modes of the cavity or the GRIN medium are also modes in a generalized sense of free space, because free paraxial propagation can be mapped onto a isotropic 2D harmonic oscillator. This is the reason why standard Gaussian modes (be them HG, LG, IG, HLG or others) maintain their intensity profile up to an expansion and corresponding decrease in intensity. Amongst other things, this means that these modes have Fourier transforms with the same shape. The new modes do not present strictly the property of self similarity under free propagation, so the profiles plotted correspond to the profile at some plane within the cavity. As it is stated in the Supplementary Information, these modes can be expressed in terms of anisotropic HGM, which accumulate different Gouy phases. Interestingly, Fig. 14 of the Supplementary Document shows that the far-

Sup. Figure 22 | Difference of the overlaps between the pump spot and the BWM and 2:1 HG modes. To illustrate the role of the misalignment in the mechanism of the mode selection we performed the following simulation. We displaced the Gaussian pump-spot (radius $250 \mu\text{m}$) by (x,y) and calculated the overlap between the pump and all BWM and HG modes up to $n=9$. Then we find, for each displacement (x,y) , the BWM mode and HG mode with the highest overlap. We plot the difference of the overlaps between the best BWM and HG mode, so that red (blue) regions indicate a higher overlap for BWM (HG) modes.

field distribution (and hence the Fourier transform) is a non-uniformly scaled version of the original mode. This property is trivial to show for the HGM in Eq. (45), and I believe that the derivation in Eqs. (43-57) shows it, but does not spell it out clearly. That is, going from the field to

its Fourier transform via a Fresnel or Fractional Fourier transform, the modes return to their same shape up to a scaling, but in between they likely take different shapes. This is hard to judge from the longitudinal plots in Fig. 14, where it is difficult to tell is the relative importance of the intensity maxima changes with z . If the modes do preserve their shape up to an anisotropic stretching, this is worth mentioning, as they would be interesting as free-space modes, not only as cavity modes. Again, even a brief mention of this behavior in the main manuscript (with more detail in the Supplementary Information) would be useful.

Our response:

We thank the reviewer for recommending that we comment more on the free-space propagation of the BWG beams. We agree that the free-space propagation behavior is an important property of this new set of modes. Therefore, we have added a new supplementary figure, Fig. S18, in which we explicitly show the BWG propagation by depicting the transverse intensity cross-sections of the BW beams at several points during their free-space propagation. Furthermore, we have added a new supplementary section, 4.3, to the supplementary materials, where we provide the exact expression for the Fourier transform of the BWG modes, namely, the far field of these modes. In this section, we have also included two new figures that display the intensity distribution of the Fourier transforms of all BWG modes up to $n=9$. Even though the resulting patterns are not simply anisotropic stretched versions of the original modes, we still observe interesting features. For example, observe the modes for $n=2$ and $n=3$, which feature unusual holes along the y -axis.

To address the free-space propagation behavior of the BWG modes in the main manuscript, we have added the following sentence:

“Note that the Bower-Wolf Gaussian modes are not scale-invariant under propagation due to their 2:1 anisotropic nature. We study their free-space propagation as well as their Fourier transform, i.e. far field, in detail in the Supplementary Material.”

Sup. Figure 10 | Fourier Transform of the even Boyer-Wolf-Gaussian modes. The intensity distribution of the first 15 Fourier-transformed Boyer-Wolf-Gaussian modes with even n .

Sup. Figure 11 | Fourier Transform of the odd Boyer-Wolf-Gaussian modes. The intensity distribution of the first 15 Fourier-transformed Boyer-Wolf-Gaussian modes with odd n .

Sup. Figure 18 | Simulation of the propagation of several Boyer-Wolf-Gaussian modes in free space. Here we show the same simulations as in Sup. Fig. (17) but as several x-y-slices along propagation.

References of potential interest to the authors. In particular the first two are directly relevant to the subject.

[1] R. Gutiérrez-Cuevas et al., Phys. Rev.A 107, L031502 (2023).

[2] M. Jaffe et al., Phys. Rev.A 104, 013524 (2021).

[3] E. G. Abramochkin and V. G. Volostnikov, J. Opt. A: Pure Appl. Opt. 6, S157 (2004).

[4] R. Gutiérrez-Cuevas et al., J. Opt. 21, 084001 (2019).

[5] M. A. Alonso and M. R. Dennis, Optica 4, 476 (2017).

Our response:

We thank the reviewer for bringing these interesting and relevant references to our attention. We have included them in the manuscript.

REVIEWERS' COMMENTS

Reviewer #1 (Remarks to the Author):

The authors have effectively addressed all of my comments, providing convincing responses. The revised manuscript is well-aligned with the expectations of the Nature Communications audience. I recommend the publication of the manuscript in its current form.

Reviewer #2 (Remarks to the Author):

The authors have addressed all my comments. The paper now seems clearer than the previous version. I have no further questions for this paper.

Reviewer #3 (Remarks to the Author):

The reviewers addressed all my comments in a satisfactory manner, so I am happy with the manuscript being published as is.

I do have two residual comments regarding the reviewer answers. Since these will have very little or no effect on the manuscript, there is no need to delay publication for them, but I would like the authors to consider them.

The reviewers addressed all my comments in a satisfactory manner, so I am happy with the manuscript being published as is.

I do have two residual comments regarding the reviewer answers. Since these will have very little or no effect on the manuscript, there is no need to delay publication for them, but I would like the authors to consider them.

The first is for their response:

"Please allow us to elaborate on our point of view. The fact that the three families of transverse modes can be written as separable functions is not only a mathematical statement but also a physical one. From all the work in integrable system it is known that **a system is separable in two-dimensions implies that it has at least two conserved physical quantities**. This means, that these solutions are eigenfunctions of two operators, one being the Hamiltonian and the other one a second order differential operator common to each family."

The statement in bold is true, but it is not exclusive. That is, other families, including the generic HLG family, are not separable, but there are also two conserved quantities: the Hamiltonian and a linear combination of the OAM and an asymmetry measure. When this linear combination includes only one or the other, we arrive at the HG or LG modes. That is, many more families of modes beyond these three are eigenstates of a second operator, and this is why they can also be selected.

Similarly, the response also mentions:

"For example, the solutions for astigmatic resonators, references [3,4] that the referee mentions, are given as summations of Hermite Gaussian modes. This corroborates our definition of these modes as fundamental, since they are the foundational modes upon which other modes are built."

Note that these modes do not need to be built as superpositions of HG modes (they can, of course). Ref. [4] gives a prescription for constructing them directly without passing through HG or LG, the formula involving HG of complex arguments.

The final comment is just a clarification. In one of my comments I mentioned:
"One basis that immediately comes to mind is that of the (anisotropic) HGM aligned with the directions of the anisotropy. This basis is not mentioned in the main body, but only in the Supplementary Information. These modes are likely naturally selected if the strength of the oscillator is detuned from the ideal 2:1."

In their response, the authors were quick to clarify that they did not modify the 2:1 ratio. I just want to clarify that I did not suggest they did, but the contrary. My comment mentioned precisely that modifying this ratio would select not the Boyer-Wolf modes but HG ones. That is, when I mention "These modes" I meant HGM, not BWM.

Again, I appreciate the work that the authors put into considering carefully all of my comments.

Observation of Boyer-Wolf Gaussian Modes Response to Comments made by Referees Second Round of Comments

We thank all three referees for recommending our manuscript for publication in Nature Communications.

Reviewer #1, #2 and #3 **accepted the paper without further questions and recommended publication in its current form.** Reviewer #3 made two residual comments that do not impact our manuscript at all, but we address them below.

REVIEWER COMMENTS

Reviewer #1 (Remarks to the Author):

The authors have effectively addressed all of my comments, providing convincing responses. The revised manuscript is well-aligned with the expectations of the Nature Communications audience. **I recommend the publication of the manuscript in its current form.**

Reviewer #2 (Remarks to the Author):

The authors have addressed all my comments. The paper now seems clearer than the previous version. **I have no further questions for this paper.**

Reviewer #3 (Remarks to the Author):

The reviewers addressed all my comments in a satisfactory manner, so I am happy with the manuscript being published as is.

I do have two residual comments regarding the reviewer answers. Since these will have very little or no effect on the manuscript, there is no need to delay publication for them, but I would like the authors to consider them.

The first is for their response:

"Please allow us to elaborate on our point of view. The fact that the three families of transverse modes can be written as separable functions is not only a mathematical statement but also a physical one. From all the work in integrable system it is known that a system is separable in two-dimensions implies that it has at least two conserved physical quantities. This means, that these solutions are eigenfunctions of two operators, one being the Hamiltonian and the other one a second order differential operator common to each family."

The statement in bold is true, but it is not exclusive. That is, other families, including the generic HLG family, are not separable, but there are also two conserved quantities: the Hamiltonian and a linear combination of the OAM and an asymmetry measure. When this linear combination includes only one or the other, we arrive at the HG or LG modes. That is, many more families of modes beyond these three are eigenstates of a second operator, and this is why they can also be selected.

Our response:

As the reviewer pointed out, our statement is true, and therefore, no further changes are necessary. It is important to note that our original statement does not explicitly claim exclusivity. Implying that we stated this and then refuting it is a logical fallacy.

Similarly, the response also mentions:

"For example, the solutions for astigmatic resonators, references [3,4] that the referee mentions, are given as summations of Hermite Gaussian modes. This corroborates our definition of these modes as fundamental, since they are the foundational modes upon which other modes are built."

Note that these modes do not need to be built as superpositions of HG modes (they can, of course). Ref. [4] gives a prescription for constructing them directly without passing through HG or LG, the formula involving HG of complex arguments.

Our response:

As the reviewer explicitly states, the modes he mentioned involve HG functions of complex arguments. This further emphasizes the fundamental role that HG beams play in the construction of laser modes, even when not explicitly built as superpositions of HG modes. The presence of HG functions in the formulation underscores their foundational importance in this context.

The final comment is just a clarification. In one of my comments I mentioned:

"One basis that immediately comes to mind is that of the (anisotropic) HGM aligned with the directions of the anisotropy. This basis is not mentioned in the main body, but only in the Supplementary Information. These modes are likely naturally selected if the strength of the oscillator is detuned from the ideal 2:1."

In their response, the authors were quick to clarify that they did not modify the 2:1 ratio. I just want to clarify that I did not suggest they did, but the contrary. My comment mentioned precisely that modifying this ratio would select not the Boyer-Wolf modes but HG ones. That is, when I mention “These modes” I meant HGM, not BWM.

Again, I appreciate the work that the authors put into considering carefully all of my comments.

Our response:

No response needed.